# Meeting Contemporary Challenges: Development of Nanomaterials for Veterinary Medicine

**DOI:** 10.3390/pharmaceutics15092326

**Published:** 2023-09-15

**Authors:** Oleksii Danchuk, Anna Levchenko, Rochelly da Silva Mesquita, Vyacheslav Danchuk, Seyda Cengiz, Mehmet Cengiz, Andriy Grafov

**Affiliations:** 1Institute of Climate-Smart Agriculture, National Academy of Agrarian Sciences, 24 Mayatska Road, Khlibodarske Village, 67667 Odesa, Ukraine; olexdanchuk@gmail.com; 2Department of Microbiology, Faculty of Veterinary Medicine, Ataturk University, Yakutiye, Erzurum 25240, Turkey; annal@atauni.edu.tr; 3European Chemicals Agency (ECHA), Telakkakatu 6, 00150 Helsinki, Finland; rochellymesquita@gmail.com; 4Ukrainian Laboratory of Quality and Safety of Agricultural Products, Mashynobudivna Str. 7, Chabany Village, 08162 Kyiv, Ukraine; dan-vv1@ukr.net; 5Milas Faculty of Veterinary Medicine, Mugla Sitki Kocman University, Mugla 48000, Turkey; seydacengiz@mu.edu.tr (S.C.); mehmetcengiz@mu.edu.tr (M.C.); 6Department of Chemistry, University of Helsinki, A.I. Virtasen Aukio 1 (PL 55), 00560 Helsinki, Finland

**Keywords:** nanotechnology, nanoparticles, veterinary medicine, diagnostics, treatment, animal disease, nanovaccines

## Abstract

In recent decades, nanotechnology has been rapidly advancing in various fields of human activity, including veterinary medicine. The review presents up-to-date information on recent advancements in nanotechnology in the field and an overview of the types of nanoparticles used in veterinary medicine and animal husbandry, their characteristics, and their areas of application. Currently, a wide range of nanomaterials has been implemented into veterinary practice, including pharmaceuticals, diagnostic devices, feed additives, and vaccines. The application of nanoformulations gave rise to innovative strategies in the treatment of animal diseases. For example, antibiotics delivered on nanoplatforms demonstrated higher efficacy and lower toxicity and dosage requirements when compared to conventional pharmaceuticals, providing a possibility to solve antibiotic resistance issues. Nanoparticle-based drugs showed promising results in the treatment of animal parasitoses and neoplastic diseases. However, the latter area is currently more developed in human medicine. Owing to the size compatibility, nanomaterials have been applied as gene delivery vectors in veterinary gene therapy. Veterinary medicine is at the forefront of the development of innovative nanovaccines inducing both humoral and cellular immune responses. The paper provides a brief overview of current topics in nanomaterial safety, potential risks associated with the use of nanomaterials, and relevant regulatory aspects.

## 1. Introduction

True dimensions of veterinary medicine are much broader than private veterinary practice focused on pets and farm animals; they reflect expanding societal needs and contemporary challenges to animal and human health and to the environment [1]. Veterinary medicine provides a crucial contribution to biomedical research, translational medical research, the study of emerging and infectious diseases, public health, production animal medicine, care of companion and exotic animals, and ecosystem management [2]. A complex of targeted activities carried out at different organizational levels characterizes the implementation of nanotechnologies in veterinary medicine. At the international level, those activities are represented as concepts, agreements, memoranda, recommendations, and other documents aimed at coordinating the development of veterinary nanomaterials in the world. At the same time, feedback from manufacturers and government agencies allows businesses to legally influence the legislative process [3].

Biomedical applications benefit significantly from the use of nanomaterials, primarily due to their size compatibility with various biological entities. Dimensions of those materials align well with the sizes of cells (1–100 μm), viruses (20–450 nm), proteins (5–50 nm), and genes (2 nm wide by 10–100 nm long). Small sizes allow nanoparticles to navigate within the body without causing a disruption to normal physiological functions [4]. Furthermore, nanoscale materials can access confined spaces that are inaccessible to larger particles, thus offering unique advantages in biomedical applications. Therefore, nanotechnology has great potential to solve many different problems in veterinary medicine and veterinary–sanitary inspection [5].

In the last three decades, rapid advances in nanotechnology have paved the way for the effective and successful application of nanomaterials in many different fields of veterinary medicine (Figure 1). The use of nanoscaled materials and drug delivery forms has proven to be effective in the diagnosis, treatment, and prevention of both infectious and non-infectious animal diseases. Those innovative approaches have demonstrated promising results in veterinary medicine, offering enhanced therapeutic outcomes and improved disease management. By leveraging nanotechnology, veterinarians can employ targeted and controlled delivery of medications that lead to improved efficacy, reduced side effects, and better overall outcomes for animals. In particular, a new generation of vaccines delivered through nanovectors showed considerably higher efficiency than conventional ones. The addition of nanomaterials to feeds enabled a direct assessment of their quality. Nanoparticle-based test systems for quality control and inspection of different products of animal and plant origin can already be found on the market [6].

In the present paper, we analyze the state of the art and outlooks for the application of nanomaterials in veterinary medicine and discuss a wide range of available nanotechnologies (medicines, diagnostic devices, feed additives, and vaccines) that are currently used to combat animal diseases. A brief overview of current topics in the safety of nanomaterials, potential risks of their use, and relevant legal aspects is also provided.

## 2. Methodology

### 2.1. Search Strategy and Selection Criteria

Electronic databases (PubMed, Science Direct, and SpringerLink) were searched for all eligible studies to date. There was no time limit for the search to ensure maximum coverage of the research on the application of nanomaterials and nanotechnologies in veterinary medicine. The following keywords were searched: “nanomaterials”, “nanotechnology”, “veterinary medicine”, “nanoparticles”, “animal health”, “vaccines”, “diagnosis”, “therapy”, “prevention”, “metabolism”, “viral diseases”, “infectious diseases”, “non-communicable diseases” of “animals”, and “domestic animals”. The keywords were used in different combinations. The identified links were imported to Mendeley reference management software. Two independent authors checked the titles and abstracts of all references. Only studies that met the article profile were included.

### 2.2. Data Extraction and Analysis

Two authors (A.L. and O.D.) performed data extraction independently. The following information was obtained for each publication: publication year, publication type (original article, review article, book chapter, patent, etc.), journal title, type of article, publication region, first author, human medicine, veterinary medicine, and animal species. The articles were conditionally divided into the following groups: diagnostics, treatment, innovative nanovaccines, applications of nanotechnology to increase productivity and prevent animal diseases, risks associated with the application of nanotechnology in veterinary medicine, and legal regulation of the nanotechnology achievements in veterinary medicine and agriculture. An article could be classified into several groups at the same time.

### 2.3. Literature Search and General Characteristics of the Studies Included

A total of 1023 records were identified, and 613 of them were excluded after initial screening by title and abstract analysis. The remaining 410 eligible records were thoroughly reviewed for the content of papers and 179 were excluded, thus leaving 231 studies reviewed in this paper. The earliest included article was published in 1988. The number of publications increased gradually in subsequent years, reaching a maximum value of 32 papers in 2021 (Figure 2).

Geographically, 32% of studies were conducted in Southeast Asia (*n* = 70), followed by 28% in Europe (*n* = 60), 24% in North America (*n* = 52), and 10% in Africa (*n* = 22), and the rest were performed in South America and Australia (Figure 3).

## 3. Current Applications of Nanomaterials in Veterinary Medicine

There is a wide variety of nanomaterials used in veterinary medicine (Figure 4). It includes nanoparticles (NPs), nanofibers, nanoplatelets, nanocapsules, carbon nanotubes, polymer nanostructures, liposomes, and micelles.

A low or absent toxicity, high biocompatibility, and capacity for biodegradation or removal by natural means are the main requirements for nanoparticles for their application in veterinary medicine [7]. According to their chemical nature, the nanoparticles may be of inorganic or organic origin. The former group includes nanoscale particles of elements (e.g., different metals) and inorganic molecules or materials (e.g., binary compounds, coordination compounds (complexes), salts, and materials) and the latter is represented by different synthetic and bio-based organic compounds, polymers, and their self-assembled aggregates (e.g., micelles, liposomes, and more complex biomolecule-based nanostructures). Regardless of their nature, NPs can be combined with different molecules (proteins, enzymes, RNA, DNA) to perform a specific function. For example, composite materials based on metal nanoparticles with hydrogels help to prevent, diagnose, and treat diseases [8].

### 3.1. Diagnostics

Due to their intrinsic physicochemical properties and biocompatibility, certain nanomaterials can find an application as probes (e.g., magnetic, fluorescent, catalytic) for different diagnostic and imaging applications. Advances in nanotechnology made it possible to develop and successfully use biochips for early diagnosis of diseases in animals. Those chips are based on hundreds or thousands of short recombinant DNA segments grafted on a silicon pool [9]. The biochips can also be used to monitor animal feeds for the presence of various pathogens. Moreover, that kind of probe is able to recognize specific organic molecules in blood, lymph, or other biological fluids and thus is able to be the principal part of nanodiagnostic devices. Bioanalytical nanosensors can recognize different xenobiotics in the bodies of farm and domestic animals [9,10,11].

Owing to their versatility and biocompatibility, nanosensor probes based on NPs of gold, silver, silica, iron oxide, europium complexes, polymers, and cadmium telluride quantum dots (fluorescent NPs) have become valuable tools for the detection of pathogens and toxins that cause animal diseases [11].

Nowadays, the majority of diagnostic laboratories increasingly rely on molecular diagnostic techniques. Integration of immune diagnostic probes with nanomaterials gave great momentum to the development of antibody-based immunodiagnostic methodologies as well as to the significant improvement of their specificity and sensitivity. Antibodies conjugated to NPs are more stable, biocompatible, and more sensitive to target antigens [12]. Effective bioanalytical kits of enzyme-linked immunosorbent assays (ELISAs) have been developed using nanomaterials to diagnose a number of animal infectious diseases, such as bird flu [13,14,15,16], post-weaning multisystem wasting syndrome (PMWS) caused by a porcine circovirus type 2 (PCV2) [17,18], Newcastle disease [19,20,21], and many others.

Various formats of nucleic acid amplification are the most frequently used molecular tests in the diagnosis of infectious diseases. Moreover, those techniques have a number of advantages over ELISA: a low cost, non-immunogenicity, and stability in response to environmental conditions [22]. Anthrax [23,24], brucellosis [25,26,27], and aflatoxicosis [28,29] have been diagnosed using nanomaterial-based molecular diagnostic kits in current veterinary practice.

Advances in nanotechnology led to the development of a variety of electronic sensors both to diagnose animal diseases and to assess the quality of livestock husbandry products. In particular, an “electronic nose” provided a fantastic possibility to detect a number of animal infections and diseases in a non-invasive way. Such sensors are available nowadays for the detection of *Mycobacterium bovis* infection in cattle [30], urinary tract infections [31], diabetes [32], and diarrhea [33], as well as for diagnosis and differentiation of upper respiratory tract infections provoked by *Staphylococcus aureus*, *Streptococcus pneumoniae*, *Haemophilus influenza,* and *Pseudomonas aeruginosa* pathogens [34]. In addition, the electronic nose technology has successfully been used for quality control of food products [35], such as olive oil [36] and milk [37].

However, the potential of molecular diagnostic tools in veterinary medicine and animal husbandry has not yet been fully revealed [30].

### 3.2. Treatment

The application of nanotechnology in medicine has great potential to restore the use of old drugs by modifying their bio-distribution, improving bioavailability, and reducing toxicity [38]. In veterinary medicine, nanoformulations enable both the dose of antimicrobials administered to animals and the drug residues in livestock husbandry products to be reduced. Highly toxic drugs, which have not been successful earlier, can now gain a second chance by being incorporated into a nanoformulated drug delivery system. The nanotechnological approach offers targeted delivery of drugs into the animal body and allows the therapeutic dose/effect ratio to be optimized as much as possible. Additionally, the outer shell of the dosage form (i.e., the nanocapsule or the NP surface coating) may also contain biologically active substances that would direct the medication towards a specific organ or tissue in the animal body [39].

Nanoparticle-mediated drug delivery provided more efficient pharmacokinetics that reduced unwanted side effects [40]. For example, the development of antibiotic resistance in microorganisms requires higher therapeutic doses of drugs to achieve the therapeutic effect. Such an increase leads to enhanced side effects of those drugs. Nanoparticle-based therapy may improve the balance between efficacy and toxicity of systemic therapeutic interventions [41]. The drug delivery could be implemented either by a strategy of passive targeting (NPs can accumulate in the target organ due to their particular size) or active targeting (using specific mediators that bind to receptors on the target cell surface) [42].

Liposomes are among the most effective systems for targeted drug delivery. They have a common structure, which provides them the possibility to contain both hydrophilic and hydrophobic drugs. Encapsulation of the active drug form in a lipid bilayer protects the former from enzymatic degradation and immunological or chemical inactivation. Therefore, liposomes both prevent the metabolism of the drug before the target tissues are reached and minimize the side effects of the encapsulated drug on healthy tissues during its circulation in the blood [43]. Drugs loaded into liposomes become bioavailable only after their release. The release rate from the liposomal vehicle could be finely tuned to achieve a desired concentration level within the therapeutic window over a period of time to have optimal therapeutic efficacy. Therefore, the effectiveness of a drug encapsulated in a liposome depends on the properties of the drug itself and on those of the liposome carrier. However, there are physicochemical properties of the liposome that determine the drug pharmacokinetics [43].

Bacterial biofilm infections are known for their tolerance to high concentrations of antibiotics and for being extremely difficult to treat. Nanotechnology-based controlled drug delivery systems demonstrated high effectiveness in eradicating bacterial biofilm-associated infections [44]. Liquid crystalline nanoparticles (LCNPs) drastically enhanced the efficacy of tobramycin in biofilm-associated infections by increasing antibiotic penetration through the biofilm matrix [45].

Nanotechnology has been widely used to deliver various biological additives, vaccines, and drugs in poultry farming [46]; e.g., silver NPs were effective against cholesepticemia in broiler chickens [47], and ZnO NPs prevented multiresistant foot dermatitis in broilers caused by staphylococci [48]. Additionally, the antimicrobial properties of zinc oxide NPs were used to treat quails [49] and to feed suckling piglets [50].

Recent studies have shown that composite materials produced by a combination of nanoparticles with nanofibers were capable of performing multiple functions (e.g., response to optical, magnetic, or pH stimuli; photo- or magnetothermal; biosensor; antibacterial; drug delivery) at the same time, which can be useful for veterinary medicine [51].

#### 3.2.1. Antibiotics and Antibiotic Resistance

Antimicrobial resistance is a global challenge recognized by several international organizations including the United Nations Organization (UNO) and the World Health Organization (WHO). Taking into consideration the growing rate of antibiotic resistance development and the long-term process of creation and introduction of new generations of antimicrobial drugs, there is a vital need for research results providing new ways to increase the effectiveness and reduce the toxicity of existing dosage forms. The importance of knowledge-based nanoparticles and nanomaterials as agents for the treatment of infectious diseases in veterinary medicine is constantly increasing [52,53].

Liposomal delivery of antibacterial drugs has offered a substantial improvement of pharmacokinetic properties; targeted, controlled, and sustained drug release; and less systemic toxicity [54]. High effectiveness of liposomal antibacterial drug forms was demonstrated for the treatment of several infectious diseases, in particular, bovine mastitis [55,56], infections caused by Gram-positive and Gram-negative bacteria [45,57], multiresistant bacterial infections [58], and *Mycobacterium avium* disease [59]. The antibacterial effectiveness of liposome-encapsulated gentamicin was proven in vivo for the treatment of intracellular infection provoked by *Salmonella enterica* serovar Typhimurium [60]. Greater efficiency and lower toxicity of enrofloxacin administered in pegylated liposomes have also been demonstrated [61]. Liposomal forms were also shown to be efficient antifungal drugs [62], especially for the treatment of canine blastomycosis [63] and fungal pneumonia [64].

Some antibacterial drugs were converted into a micellar form, thus gaining a second life. Drinking is the most convenient way to administer drugs in poultry farming. However, it could be problematic to administer fat-soluble substances including a large number of antibiotics. Conversion of antibiotics into micellar forms improves their bioavailability and pharmacokinetic parameters. For example, micellar tilmicosin demonstrated high efficacy in broiler chickens [65]. Blastomycosis in dogs was effectively and safely treated with Amphotericin B lipid complex [63].

An alternative to antibiotics is vitally needed nowadays. Metal nanoparticles offer one of those alternatives since microorganisms possess no genetic basis to counteract those NPs. Therefore, they have widely been used to target bacterial resistance mechanisms like that of *Escherichia coli* [66]. For example, silver nanoparticles (AgNPs) are well known for their efficacy against antimicrobial resistance [67], and the toxic effects of AgNPs in animals are also well studied. Therefore, silver nanoparticles have been used in pig farming as an antimicrobial additive for piglets [68] and an antiviral agent against African swine fever virus [69]. In 2014, the first attempts were made to establish the effectiveness of gold, silver, copper, and platinum nanoparticles as antimicrobials for the treatment of bovine mastitis [70]. Since then, a number of efficient commercial preparations containing metal nanoparticles have been developed for the treatment of diseases [71]. Those nanoformulations demonstrated a high antimicrobial efficacy against *E. coli*, *Streptococcus uberis*, *Staph aureus*, *Candida albicans*, and *C. krusei*. Metal NPs have also demonstrated a promising activity against biofilms and antibiotic-resistant bacteria [72]. In particular, the AgNPs were highly effective against antibiotic-resistant bacteria [73] and biofilms [74]. The antibiofilm activity mechanism of AgNPs and Zn-calcium silicate NPs was proposed [74].

Zinc oxide nanoparticles were demonstrated to have high antibacterial efficiency in the treatment of blood mastitis [75]. Those NPs were also reported as antimicrobial feed additives for animals [76].

The nanotechnological approach made it possible to enhance the effect of known antibacterial agents; e.g., it was established that the nanoencapsulated form of curcumin minimized some detrimental effects caused by *L. monocytogenes* in gerbils in terms of tissue damage and interference with energy metabolism enzymes [77].

In aquaculture, up to 80% of antibiotics administered in the form of granulated medicated feed accumulate in the aquatic environment [78]. Several nanotechnological alternatives to conventional antimicrobial agents used in aquaculture have increasingly been suggested [79], particularly those containing metal NPs [80]. A promising approach involving environmentally friendly carbon dot-TiO_2_ nanocomposites was proposed for the removal of antibiotics from aquaculture effluents through solar irradiation [81]. When activated by light, the TiO_2_-NPs exhibited bactericidal effect against several fish pathogens (*Streptococcus iniae*, *Edwardsiella tard*, and *Photobacterium damselae*) [82], and ZnO-NPs were effective against *Vibrio harveyi* [83].

#### 3.2.2. Antiparasitic Properties of Nanoparticles

Nanotechnology offers several efficient alternatives for the treatment of parasitic diseases of both animals and humans. In particular, AgNPs obtained from the *D. flagrans* fungus showed a high nematicidal efficiency, since the NPs were able to penetrate the cuticles of the larvae and subsequently provoke the death of zoonotic nematodes [84]. ZnO [85] and silver [86] nanoparticles were effective against coccidiosis in rabbits. Hydrosols of nanoparticulated hydrated antimony (V) and bismuth (III) oxides were proposed as new dosage forms for topical treatment of cutaneous leishmaniasis provoked by several species of *Leishmania* parasites (*L.amazonensis*, *L.brasiliensis*, and *L. guayanensis*). The therapeutic potency of the NPs was higher with respect to molecular forms of pentavalent antimonial drugs (e.g., Glucantime^®^ and Pentostam^®^), and the nanohybrids exhibited a higher efficacy at doses 2–3 times lower [38,87]. The possibility to apply the nanoformulations topically provided an opportunity to develop a patient-friendly treatment for cutaneous leishmaniasis instead of the current systemic course of IM injections.

Highly efficient liposome-based antiparasitic vaccines were developed. The administration of a DNA (encoding the MIC3 protein)-containing plasmid encapsulated in liposomes to sheep elicited a significant and effective immune response against *Toxoplasma gondii* [88].

#### 3.2.3. Tissue Scaffolds (Electrospun)

The manufacturing of nanofibers by electrospinning provided the possibility to fabricate scaffolds used for tissue reconstruction and regeneration. For example, biodegradable poly(lactide-co-glycolide)/hydroxyapatite (PLGA/HAp) electrospun nanofibers were used as bone scaffold biomaterial to repair critical-sized segmental bone defects in a canine model [89]. Nanofibers have also been applied as scaffolds to support tissue-specific cell functions and tissue-mimicking systems for tissue/organ regeneration [90].

Stem-cell-based cell therapies have shown promising results for novel biomedical treatments of various diseases, including myocardial infarction, hind limb ischemia, and stroke. Intramyocardial injection of a hydrogel containing bone-marrow-derived stem cells encapsulated in α-cyclodextrin/poly(ethylene glycol)–b-polycaprolactone-(dodecanedioic acid)-polycaprolactone–poly(ethylene glycol) into the infarcted myocardium of a rabbit increased the survival and retention of transplanted cells and further improved the impaired cardiac function compared to BMSC implantation alone [91].

#### 3.2.4. Administration of Nanoformulations in Neoplasms

In recent decades, much research has been dedicated to nanoparticles capable of detecting and destroying cancer cells. Nanopharmaceuticals involve loading active pharmaceutical ingredients into nanocarriers in order to improve the solubility and bioavailability of the former, to extend their half-life, to enhance the pharmacokinetic properties, to modify the release profile, to reduce acute and chronic toxicity, and to achieve site-specific targeting [92]. In veterinary medicine, the nanoformulation approaches encompass all the different delivery systems shown in Figure 4 and enable the targeted delivery of antitumor drugs to reduce or avoid the side effects of conventional chemotherapy [93]. On one hand, nanoparticle-based drugs overcome several limitations associated with conventional diagnostic and therapeutic protocols in veterinary oncology owing to their intrinsic size-related properties, favorable pharmacokinetics, tumor-targeting properties, and consequent superior efficacy and toxicity profiles [94]. On the other hand, nanomedicine faces several scientific challenges in terms of unclear toxicity and bio–nano interactions, reproducibility and transparency problems, and the complexity and costs of nanoparticle manufacturing [95]. A detailed discussion of several important nanoparticle-based formulations and principal tumor targeting mechanisms (EPR (passive) and targeting (active) of cell surface receptors) was provided in [94].

An analysis of the interactions between 35 types of nanoparticles and nanocarriers and almost 500 types of tumor cells revealed thousands of biological features that determine the ability of different cells to absorb different types of NPs [96]. Those data opened up new opportunities for the use of nanoformulations in tumor treatment.

Nanomedicines achieve tumor-targeted delivery mainly through the enhanced permeability and retention (EPR) effect [97]. Nanocarriers not only improve the delivery of drugs but also significantly change their pharmacokinetics, leading to a decrease in toxicity and side effects [98]. For example, studies using animal models demonstrated the high efficiency of dextran-coated iron NPs conjugated to antagomirs for the treatment of metastatic breast and brain cancers [99] as well as a triple action (CXCR_4_ antagonism and downregulation of miR-210/KRAS^G12D^) and significant improvement of the delivery of cholesterol-modified paclitaxel NPs to pancreatic cancer cells [100,101]. Spontaneous tumors in cats and dogs have been proposed as the best animal models for human cancer and thus have been used in preclinical studies for the development of new drugs and imaging probes [102].

Recently, a novel immunological approach to cancer therapy has emerged with the development of single-domain antibodies (sdAbs) or nanobodies, which were engineered from heavy-chain-only antibodies present only in camelids and sharks [103]. Small size and low molecular mass (12–15 kDa vs. 150–160 kDa in common antibodies), high stability, strong antigen binding affinity, water solubility, and natural origin made them an excellent choice for the development of next-generation biopharmaceuticals [103,104].

However, there is an essential question of to what extent the therapeutic practice of human oncology can be applied in the veterinary one, since both approaches are largely similar. Various types of cancer, such as squamous cell carcinoma (SCC) of the oral cavity, mammary gland carcinoma, osteosarcoma (OSA), and transitional cell carcinoma, which are very similar to human tumors spontaneously develop in companion animals (e.g., cats and dogs) [102]. For example, mammary gland tumors in cats and dogs have epidemiological, clinical, morphological, and prognostic characteristics that are very similar to those of human breast carcinoma. Therefore, cats and dogs are considered an excellent model for studying hormone-independent breast cancer in humans [105]. Thus, it is absolutely logical to find reports on the use of gold nanorods in photothermal therapy of mammary tumors in cats [106].

Certainly, the adaptation of treatments and techniques is quite a long process, and it would be necessary to take into account the species, breed, and individual characteristics of the animals. For example, paclitaxel is a highly effective drug for the treatment of many types of human cancers, but it cannot be applied universally to dogs, due to their high sensitivity to the medication. However, a nanoformulated composition CTTI 52010 containing paclitaxel was found to be well tolerated by dogs with tumors in a phase I/II study when the drug was administered IV with a starting dose of 80 mg/m^2^ [107].

Investigations of the therapeutic effects of different metal nanoparticles (e.g., Au, Ag, Pt, and Fe) on animal tumors are quite common [108]. In 2015, Feldhäusser et al. successfully applied mitochondria-targeted Pt(IV) nanoparticles (T-Platin-M-NPs) to treat brain tumors in dogs [109]. The conjugation of doxorubicin (Dox) to 4 nm Au-NPs stabilized with glutathione (Au-GSH-Dox) enhanced the antiproliferative activity and cytotoxicity of the drug in Dox-resistant feline fibrosarcoma cell lines [110]. Interestingly, a polydisperse colloidal solution of gold (i.e., a mixture of AuNPs of different sizes between 10 and 50 nm) at a final concentration of 10 μg/mL inhibited the growth of LNCaP human prostate cancer cells, while a solution of monodisperse 20 nm AuNPs had no effect. The polydisperse AuNP solution also stopped the growth of human PCa xenotransplants in mice upon parenteral administration in the dose range of 0.64–6.4 μg/kg body weight [111].

A nanoformulation of hyaluronan modified with *cis*-diamminedichloroplatinum(II) (cisplatin) and paclitaxel demonstrated promising treatment results for oral melanoma, oral sarcoma, and anal gland adenocarcinoma in dogs [112].

Nanopreparations can overcome current limitations in cancer treatment and provide targeted drug delivery to mitochondria, improving the pharmacokinetic properties and bio-distribution profiles of the active substances [113]. Enzymatic delivery of magnetic NPs into the mitochondria of live cells represented an innovative strategy for the application of targeted drugs in biomedicine and cancer therapy [114].

Metal nanoshells obtained on a dielectric silica core possess tunable optical properties. Their optical resonances, and thus a strong light absorption, may be tuned to occur in the near-infrared region. That property paves the way to thermal ablative therapy for cancer. For example, silver nanoshells on a silica surface treated with PEG-monothiolate induced photothermal tumor cell damage in vitro upon exposure to coherent NIR light at 820 nm (35 W/cm^2^) [115]. The ability of superparamagnetic nanoparticles based on iron oxides to penetrate tumor cells is improved under the influence of an external magnetic field for targeted drug delivery [9]. Moreover, inorganic metal-based NPs may be directly visualized by electron microscopy, while organic NPs need to be labeled with fluorescent dyes or radioisotopes in order to become visible [116].

In summary, nanoformulations have found a wide range of applications in neoplasms; however, it is currently impossible to distinguish the development of veterinary and human approaches to cancer treatment using nanotechnology [117]. The period of knowledge accumulation in this direction involves the widespread use of animal models to understand general mechanisms [118]. Thus, companion animals with spontaneous oncological diseases are increasingly recognized as animal models with great potential for human oncology research. Many anticancer nanodrugs have recently been developed [98]; among them are nanovaccines, which represent an important strategy for the prevention and treatment of tumors and are discussed in Section 3.3.

#### 3.2.5. Gene Therapy

As mentioned above, the dimensions of nanomaterials are comparable to those of nucleic acids, and therefore, NPs are regarded as promising gene delivery vectors. Obviously, the main research efforts in the area of gene therapy are focused on treatment of human diseases. However, animal models are widely used to study the efficiency, pharmacodynamics, and pharmacokinetics of nanoformulations [119]. Those results enable the application of the latest achievements of gene therapy in veterinary medicine as well. Currently, the main animal models are mice, hamsters, guinea pigs, rabbits, cats, dogs, pigs, and cows [120]. The therapeutic use of messenger RNA (mRNA) has given rise to great hopes for the development of treatment for a wide range of incurable diseases. Recent rapid advances in biotechnology and molecular medicine have made it possible to produce almost any functional protein/peptide in the animal/human body by introducing mRNA as a therapeutic agent or vaccine [121].

Due to the use of large animals as experimental models in gene therapy, veterinary medicine is becoming an increasingly important translational bridge between preclinical research and human medicine [122]. Large animal models have many advantages over small ones [123,124]. For example, those species and humans share many anatomic and physiological similarities, similar living environments, and environmental risk factors for the development of certain diseases [122]. Additionally, the metabolism intensity is quite high in small laboratory animals. Thus, it may not always be correct to compare the effect of certain substances on their body with that on the human one, while the metabolism in large animals is quite similar to that in humans in terms of intensity. Moreover, pigs are the best analogs to humans in terms of body weight, which is useful for dosage development. Different gene therapies were developed for the treatment of arthritis and infectious and cardiovascular diseases using pig and horse models [125]. Sheep were used to develop gene transfer methodologies and for gene-marking studies. The efficiency of gene therapy for the treatment of hereditary hypercholesterolemia [126] and heart diseases [127] was proven in rabbits. The feline brain is well developed both functionally and physiologically, and therefore, cats are of particular interest in the development of gene therapies for certain neurological disorders [128]. Furthermore, successful gene therapy approaches were also reported for fibrosarcoma and hereditary and heart diseases [122]. Currently, therapeutic gene transfer is successfully used to treat cardiovascular diseases in dogs and cats [129]. Over 50% of genetic diseases in dogs are caused by mutations in the same genes as in humans, and their immune system is remarkably similar to that of ours. Positive results of genetic treatment of mucopolysaccharidosis VII [130], hemophilia A and B [131], and malignant melanoma [132] in dogs were reported. A method of gene therapy for melanoma was successfully tested in horses [133]. The above results obtained in large animals demonstrate the high efficacy and safety of gene therapy, which was well tolerated. Therefore, further research is strongly encouraged in that field of veterinary medicine, bearing in mind a subsequent application of the therapies developed in human clinical trials.

### 3.3. Innovative Nanovaccines

Vaccines are one of the most important public health tools that play an important role in prophylaxis and treatment of infectious diseases. In turn, nanotechnology is an integral part of nanotechnology, biotechnology, information technology, and cognitive science (NBICS) technologies [134]. The development of modern vaccines cannot be imagined without digital technologies, advances in biotechnology, cellular engineering, and several other industries that are directly or indirectly related to nanotechnology [135]. Nanomaterial-based vector systems exhibit low immunogenicity, easily adjustable molecular weight and structure, and easy conjugation of the functional moiety to the nanomaterial backbone [136].

The global market of veterinary vaccines is expected to grow from USD 7 billion in 2020 to USD 10.18 billion in 2025 with an average annual growth rate of 7.1% [137].

Veterinary medicine is in the avant-garde of the development of innovative third-generation nanovaccines containing DNA, RNA, and recombinant protein components that induce both humoral and cellular immune responses [138]. Advances in nanotechnology enabled the assembling of complex nanoparticles possessing adjuvant properties for a new generation of synthetic vaccines containing antigens of different molecular sizes such as proteins, peptides, and oligosaccharides. Ester-bonded pseudo(polyamino acids) are new polymers that are harmless to the body and can be used as adjuvants in vaccine development [139]. The first synthetic vaccine particle (SVP) was introduced in 2012 [140]. SVP NPs are able to mimic different antigens and trigger the immune response; this is a novel nanotechnological avenue in vaccinology.

DNA nanovaccines show significantly better results in veterinary practice than classical vaccines, but the protection efficiency provided by naked plasmids is usually between 28% and 90% [135]. DNA delivery systems based on metal or polymer nanocarriers could resolve the problem; e.g., pegylated AgNPs for vaccine gene delivery were reported [141].

These NPs penetrate the target cell membrane through endocytosis and release the DNA vaccine into the cytoplasm. The negatively charged cell membrane is a barrier to large polynucleotides like DNA bearing the same charge; the problem may be solved using a polycation-based delivery system [142]. Moreover, nanoscaled cationic polymer particles increase the chemical stability of DNA vaccines and thus induce an enhanced immune response [143]. A vaccine encapsulated into polyethyleneimine (PEI) NPs is able to activate both humoral and cell-mediated immunity after vaccination [144]. An IBDV DNA vaccine with a PLGA-PEI nanocarrier demonstrated higher efficacy (up to 80%) in reducing morbidity and mortality of birds. This efficacy value is significantly higher than that of similar classical vaccines [145]. Nanovaccines based on a liposome-encapsulated plasmid were also developed against avian influenza (AIV) in chickens [146] and Anatid herpesvirus 1 in ducks (a liposome-encapsulated plasmid-chitosan) [147], although the details regarding their protective properties were not disclosed.

A plasmid-based vaccine for turkey sweat coronavirus (TCoV), which uses a disulfide-crosslinked low-molecular-weight linear polyethyleneimine (CLPEI) carrier, offered a certain protection from TCoV infection by triggering a humoral immune response that can mitigate or eliminate symptoms and decrease the viral load [148]. A AuNP-encapsulated DPV vaccine elicited a stronger humoral immune response but a weaker cell-mediated immune response in ducks [149]. Vaccines based on chitosan NP carriers protect the DNA from degradation; e.g., effective vaccines against both the Newcastle disease virus (NDV) of birds [150] and *Pasteurella multocida* [151] were developed by encapsulation of DNA in chitosan NPs. However, despite all the advantages of chemical DNA nanocarriers, biological vectors like *Salmonella* (70%) and LAB (20%) are still widely used for oral delivery. At the moment, only around 10% of vaccines are nanoformulated [135].

Due to their accuracy, safety profile, and flexible production, mRNA vaccines are gaining popularity as a new alternative to conventional ones [152]. Vaccines against infectious salmon anemia were developed using an inactivated ISAV virus encapsulated in chitosan NPs; they demonstrated a high protection level exceeding 77% [153]. Chitosan-coated poly (lactic co-glycolic acid) nanoparticles were developed as an effective and safe mucosal immune delivery system for an NDV DNA vaccine. The nanoformulated plasmid NDV F-gene vaccine pFDNA-CS/PLGA-NPs induced greater cellular, humoral, and immune responses in chickens when compared to the single plasmid vaccine [154].

The data on nanoformulated vaccines for poultry and their effectiveness are shown in Table 1.

Advances in molecular biology, RNA technology, vaccinology, and nanotechnology have led to the development of a range of mRNA-based therapeutics [121]. The most promising of them are the mRNA-based vaccines against infectious diseases, which have a number of advantages over classical vaccines. Modified mRNA vaccines are easy to manufacture, safe, and highly immunogenic. Stabilized lipid-based nanoformulations protect the mRNA from enzymatic degradation [155]. Primarily, such vaccines were developed for human medicine, but they had been tested on animals theretofore. This fact creates prerequisites for their use in veterinary medicine as well. One particular example includes the mRNA vaccine developed against Lyme disease, which has recently become increasingly widespread among humans and domestic animals in the Northern Hemisphere. The vaccine showed high efficiency in guinea pig tests by inducing tick resistance and preventing the transmission of *Borrelia* spp. [156,157]. EBOV envelope glycoprotein-based mRNA vaccines demonstrated high efficacy and elicited sustained immune responses and protection in guinea pigs against the Ebola virus disease [158]. FMD virus vaccination using NP-conjugated peptides induced sustained immune and humoral responses in sheep [159].

**Table 1 pharmaceutics-15-02326-t001:** Nanovaccines for poultry and their effectiveness.

Host/Pathogens	Vehicle	Target Antigen/ Vaccination Route	Immune Responses	Protection/Ref.
Chicken/AIV	pHEMA	H6/IM	Ab response	Reduced virus shedding/[160]
Turkey/ *C. psittaci*	Branched PEI	OmpA/IM	IgG and increased CD4/CD8 rate response	Reduced *C. psittaci* shedding, shortened clinical sign period/[161]
Chicken/NDV	Lipofectin	F and HN/IM	Anti-F Ab	80%/[162]
Chicken/NDV	Chitosan NPs	F/IM/NAS	IgA/IgG and lymphocyte proliferation	IM: 80%/[163]
				IN: 100%/[163]
Turkey/TCoV	Naked plasmid + PEI and sodium hyaluronate	4F, 4R/IM	Anti-TCoV S Ab and VN titer	Decrease in clinical signs from 5/5 to 1/5 or 2/5/[148]
Chicken/NDV	Nano-chitosan	F/IM/NAS	IgG and IgA and lymphocyte proliferation	80% (IM); 100% (i.n.)/[164]
Egg embryonation/IBDV	Naked plasmid/killed vaccine	VP2, VP3, VP4 + killed virus booster/IO/IM	Anti-IBDV Ab and lymphocyte proliferation	100%/[165]
Chicken/IBDV	Poly lactic-co-glycolic acid (PLGA)	VP2/IM, PO, OU	Stimulation of CD4 and CD8 T cells, high level of IgG	80%/[145]

The development of lymphoid tumors in cattle is the most prominent clinical manifestation of BLV infection [166]. The virus causes significant economic losses worldwide due to its high prevalence and a lack of effective treatment. Vaccination is the only effective way to fight the virus. However, the low efficacy of traditional vaccines is probably due to inadequate or short-lived stimulation of all immunity components [167]. Therefore, the development of new generations of vaccines incorporating all the advantages of nanotechnology is currently a highly relevant issue. Viral peptides encapsulated in mannan-coated liposomes as a delivery system induced a significant humoral response and specific Th1-type immunity in mouse and sheep models [168]. To protect susceptible cattle against BLV, a peptide vaccine was developed using 3D modeling and nanotechnology [169].

Injection-free ways of vaccine administration offer the possibility of mass vaccination, low cost, and high efficiency. Various nanobiomaterials such as mucoadhesive polymers, lipids, and polysaccharides were used for antigen delivery, leading to improved production of antibodies in the vaccinated animals [170]. Table 2 shows nanoformulated vaccines for veterinary use administered by a respiratory route.

### 3.4. Application of Nanotechnology to Increase Productivity and Prevent Animal Diseases

Recently, agricultural applications of nanotechnology have received more and more attention, particularly as a tool to increase the productivity of domestic animals.

Nanotechnology is a promising tool to enhance the bioavailability of fat-soluble organic substances; e.g., nanoformulated micellar forms of vitamins were much more effective than any other aqueous analogs [54]. Administration of fat-soluble vitamins in the form of micelles provides the possibility to administer the dose with drinking water, reduce costs, and enhance efficiency at the same time. For example, oral supplementation of the micellar form of vitamin E in horses increased the vitamin concentration in the blood and provided support to the antioxidant protection system during intense training [182]. In weaned piglets and adult pigs, lower oral doses of micellar α-tocopherol showed a better effect than any other form of the vitamin [183]. Feeding a julep containing the micellar form of vitamin E to fattening piglets and pigs prevented the consequences of technological stress and increased activity of the antioxidant protection system and general resistance of the animals [184].

The intensive industrial technology of animal husbandry often requires a high dosage of inorganic salts as mineral feed additives, and these inorganic salts lead to environmental pollution because of their low bioavailability [185]. Nanoparticles of inorganic substances are highly bioavailable and produce the same effect at much lower doses when compared to the corresponding mineral forms [186]. Therefore, the administration of smaller doses of metal-containing NPs could be a potential alternative solution to many problems and has great application potential. Several metal NPs are currently used successfully to improve animal and plant productivity [187,188].

Nanoparticles containing Zn, Ag, Cu, Au, Se, Cr, Ca, Mn, or Co have become the most widespread in animal husbandry. The diagrams in Figure 5 and Figure 6 represent an analysis of scientific publications in the PubMed, Science Direct, and SpringerLink databases in the period from 1990 to 2022. While Ag and Au NPs are more widely used as feed preservatives and growth promoters to replace antibiotics, Zn, Cu, Se, Cr, Mn, and Co are essential trace elements making part of several main enzymes essential to stimulate metabolism and improve animal resistance and productivity. Analysis of scientific publications devoted to the application of inorganic NPs in animal husbandry by species indicated that more than 60% of papers dealt with poultry farming (with the main focus on meat (35%) and egg (27%) production), up to 20% of them dealt with pig breeding, and about 10% were dedicated to cattle breeding.

In our opinion, such a distribution correlates directly with the intensity of the livestock production technology. On one hand, the highest technological intensity in poultry meat and egg production promoted the use of innovations that provide a rapid economic effect. That is the stimulus for increasingly growing research efforts in this area. On the other hand, much more care is needed to make pig and cattle breeding profitable. Technologies for keeping cattle and pigs are rather conservative, especially for small farms; thus, the implementation of the latest innovations could be a challenge and lags far behind.

Nanoparticles of biogenic elements are widely used in industrial poultry farming [189]. For example, gold NPs upgrade growth and immune parameters [190]. Selenium NPs affect the physiological responses, immune status, and productivity of broiler chickens [191]. Calcium nanoformulations improve the physical and biological characteristics of eggshells [192]. Silver nanoparticles and starch-AgNP nanocomposites were successfully used in poultry as a growth-promoting feed additive and as a safe alternative to antibiotics [193,194,195]. Zinc NPs improve broiler chicken performance [196], the immune response, and the quality of eggs [197] and increase the quality of broiler chicken meat [198]. It is interesting to compare the properties of a mineral zinc oxide feed additive with those of a nanoformulated one. ZnO-NPs were successfully used to increase broiler productivity at a dose of 90 mg/kg diet, while a conventional mineral zinc oxide gave the same effect at a dose of 3000 mg/kg diet. Both supplements did not significantly affect blood parameters and had the same antibacterial activity against *E. coli*. Additionally, the ZnO-NPs enhanced the immune response and antioxidant defense of broiler chickens [199], while the mineral zinc oxide was found to provoke dose-dependent toxicosis [200].

In pig farming, the application of nanoformulated essential microelements has attracted increasing attention as a means to stimulate animal metabolism and resistance. For example, Zn-based NPs were used to fortify weaning piglets and to improve their digestion and productivity [185,201,202], especially as an alternative to the mineral zinc oxide [203]. In particular, the administration of nanoformulated Mg, Zn, Ge, and Ce feed additives to pigs increased the activity of their antioxidant defense system, stress resistance, and productivity [184].

In cattle breeding, the application of nanoformulated selenium and zinc enhanced animal resistance and improved milk production capacity [204]. The addition of Zn-based NPs to the diet led to an increase in performance, rumen fermentation, and antioxidant system activity in calves [205] and goats [206]. Selenium NPs increased selenoprotein (Sel) gene expression and the selenium concentration in the milk of lactating dairy cows [207] and improved the feed assimilation and rumen fermentation in sheep [208].

The administration of copper NPs to rabbits stimulated their growth [209], while AgNPs improved their performance and antioxidant status [210].

The main obstacle to the widespread adoption of NPs in aquaculture is their high cost [211]. However, nanotechnologies are increasingly being used in the delivery of dietary supplements [212]. Chitosan NP supplementation enhanced the survival, growth performance, and meat quality of Nile tilapia (*Oreochromis nilotica*) [213]. SeNPs were employed as natural antioxidants and metabolic regulators in aquaculture [214].

## 4. Risks and Hazards of Nanotechnology Applications in Veterinary Medicine

Nowadays, safety issues in different applications of nanomaterials are highly relevant. In the Parma Declaration on the Environment and Health Protection, the representatives of the European WHO member states called for intensifying the research on the potential harmful effects of nanomaterials on human health [215]. Between 2009 and 2018, potential risks were identified for 130 nanomaterials and for more than 400 types of nanotechnological products and their production technologies. Particular attention has to be paid to an increasing accumulation of nanoparticles in the environment. Interdisciplinary research in those areas has become increasingly important [216]. Nanoparticles are everywhere, from mobile phone cards to nanofertilizers and nanoformulated feed and food additives. Therefore, NPs might represent a new class of potentially toxic substances capable of affecting the ecosystem. Consequently, much effort is needed to evaluate the long-term effects of NPs on both individual organisms and the ecosystem [217].

NP-containing disinfectants, medical preparations, biomaterials, vaccines, and immunobiological agents are widely used in veterinary medicine. All those require the development of new control and testing methodologies, particularly for the detection of the cyto-, geno-, and ecotoxicity of nanomaterials [217].

Studies on the impact of NPs on animal bodies revealed a number of different dose-dependent side effects, such as a pro-oxidant effect, the ability to cause inflammation, oxidative stress, and the modification of mitochondrial distribution [218]. For example, CuNPs caused a proliferation of brain capillary endothelial cells in rats, even at low concentrations [219]. Exposure of mice to AgNPs affected the blood–brain barrier and was accompanied by neurotoxic effects [220]. Similar results were also obtained in experiments on pigs; moreover, the cytotoxicity of AgNPs was found to increase as the particle size decreased [221]. Silver nanoparticles may also induce oxidative stress, apoptosis, and the disruption of steroidogenesis in bovine granulosa cells [222]. Cadmium-based quantum dots (QD705) induced persistent inflammation and formation of granulomas in mouse lungs; pegylation of those NPs did not prevent the side effects [223,224].

Ag, ZnO, or CuO NPs are often used as bactericides in veterinary medicine; they may also have toxic effects on non-target organisms when released into waste and the environment [217].

It was demonstrated that cationic liposomes used as DNA vaccine delivery vectors may induce nonspecific inflammation and allergic reactions [225].

The toxic impact of nanomaterials on aquatic organisms has been extensively studied and reported by numerous researchers [211]. To ensure sustainable development and management of nanotechnologies in aquaculture, a thorough analysis of potential risks to the environment and human health is of crucial importance [226].

Nanomaterial-based diagnostic tests specific to different animal pathogens still have not achieved balanced sensitivity, specificity, reproducibility, and cost-effectiveness [11].

Therefore, the expanding production of nanomedicines and feed additives requires good manufacturing practices (GMPs), establishment of regulations, and further improvement of technological safety and efficiency [227]. Since the results on the toxic effects of NPs are often contradictory, further research on their effect on individual tissues, genetic material, and the immune system of the animals is vitally needed.

## 5. Legal Regulation of Nanomaterials Implemented in Veterinary Medicine and Agriculture

Nowadays, there are several control systems for the use of nanomaterials (e.g., Control Banding Nanotool, IVAM Technical Guidance, Stoffenmanager Nano, ANSES CB Tool, NanoSafer, and Precautionary Matrix), which were created to provide support to the manufacturers and regulatory agencies in assessing potential hazards associated with nanomaterials [228]. However, all of them were based on different concepts and have different input parameters and output formats, and therefore, it has not been possible to immediately combine those different models into a larger coherent framework [229]. There is a primary need to harmonize those different systems of risk assessment and control over the use of nanomaterials, which will allow a unified system that can form the basis of legal and regulatory acts.

In the USA, the production, circulation, and use of veterinary medicines and feed additives are regulated by the legislation of each individual state. However, significant work has been conducted to unify those provisions. The EU countries signed the “Community Code on Veterinary Medicinal Products”, which came next in this direction [230]. It should be noted that the use, circulation, and production of veterinary nanopreparations and NP-containing feed additives still fall under the legislation developed for human medicinal products. No veterinary medicinal product may be placed on the EU Member State market without authorization issued in accordance with the EU Directive [230] and Regulation (EC) No. 726/2004 [231].

Since a large number of nanoformulations are based on well-known active pharmaceutical ingredients that were given a new dosage form (e.g., encapsulated forms of antibiotics), admission of those nanomedicines to the market does not require proper clinical and other tests. The issue with feed additives and medicated feeds is more complicated since medicated feeds were not mentioned in the Directive [230].

Whereas the number and areas of nanotechnological applications in veterinary medicine are dynamically developing, there is a vital necessity for legal regulation of the production, circulation, and use of veterinary nanoformulations and feed additives.

## 6. Prospects for Nanotechnology in Veterinary Medicine

Nanotechnology has emerged as a promising field having a variety of applications, including those in veterinary medicine. Recent advancements in nanotechnology have paved the way for innovative strategies in the diagnosis, treatment, and prevention of animal diseases [5].

The development of enhanced drug delivery systems is one of the most significant contributions of nanotechnology to veterinary medicine [39]. That approach may help to address the antibiotic resistance issue in animals and contribute to the development of more effective treatment options. Nanoformulations have demonstrated improved efficacy against parasites, providing a potential solution for the control and treatment of those infections in veterinary practice [52,53]. Moreover, nanotechnology paves the way for targeted therapies for neoplastic diseases, although more research and development are needed in this area.

Nanomaterials have been explored as gene delivery vectors in veterinary gene therapy [119]. Due to their size compatibility, nanoparticles can efficiently deliver therapeutic genes to target cells and tissues [122]. That strategy holds great potential for the treatment of genetic disorders and other diseases with a genetic component in animals. Ongoing research in this area aims to improve the efficiency and safety of gene delivery systems for veterinary applications [120].

Nanotechnology has enabled the development of innovative veterinary nanovaccines [138]. Nanoparticles can enhance the immune response by delivering antigens directly to immune cells and triggering both humoral and cellular immune responses [144]. This approach may lead to improved vaccine efficacy and protection against infectious diseases in animals.

While nanotechnology offers many exciting possibilities, it is also important to consider the safety aspects associated with the use of nanomaterials in veterinary medicine [217]. Ensuring the safety and regulatory compliance of nanomaterials will remain a crucial aspect of their integration into veterinary practice.

Thus, nanotechnology holds significant promise for advancing veterinary medicine. Future achievements in the field may involve further research on targeted therapies, the development of nanovaccines against emerging infectious diseases, and continued exploration of gene delivery systems [52].

## 7. Conclusions

A wide range of nanotechnologies has been implemented in veterinary medicine, including pharmaceuticals, diagnostic devices, feed additives, and vaccines. Drug delivery mediated by nanoparticles ensured more efficient pharmacokinetics and enabled the restoration and use of some known toxic drugs due to the modification of their bio-distribution, improvement of bioavailability, and reduction in unwanted side effects. For example, the applications of nanoformulated antibiotics demonstrated their higher efficacy and reduced toxicity, providing a potential solution to overcome antibiotic resistance.

Different inorganic and organic nanoparticles have been widely used for the treatment and prevention of various animal diseases. Nanoformulations are currently being used as therapeutics, feed preservatives, growth stimulants, and antibiotic replacements. At the same time, efficient NP-based antitumor drugs tested on animals still have not been widely implemented in veterinary medicine owing to their very high cost.

The veterinary vaccine market has been rapidly advancing, with nanotechnology finding a broad implementation in the field. Innovative nanovaccines have been capable of inducing both humoral and cellular immune responses. The third-generation vaccines manufactured using nanotechnology offered a range of undeniable advantages over their traditional counterparts. Hence, they are increasingly used in animal husbandry and poultry farming in particular.

The wide application of multi-component nanoformulations in veterinary medicine and agriculture may raise a number of relevant questions and have complex consequences that require further in-depth and detailed research. Veterinary medicine also needs to focus on the prevention of pathological conditions (symptoms, syndromes, diseases) associated with the incorrect use of nanotechnological products in agriculture and the accumulation of nanoparticles in the environment.

And last, but not least, the development of a solid legal regulatory basis for the production and application of nanoformulations as veterinary drugs, feed additives, growth promoters, and vaccines is vitally needed.

## Figures and Tables

**Figure 1 pharmaceutics-15-02326-f001:**
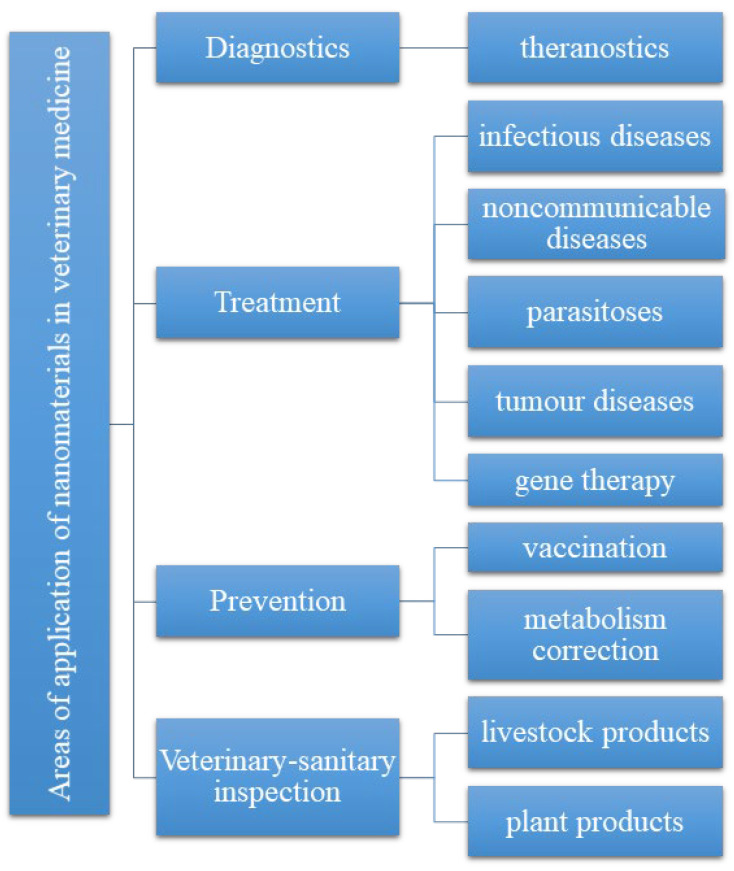
Main application areas of nanomaterials in veterinary medicine.

**Figure 2 pharmaceutics-15-02326-f002:**
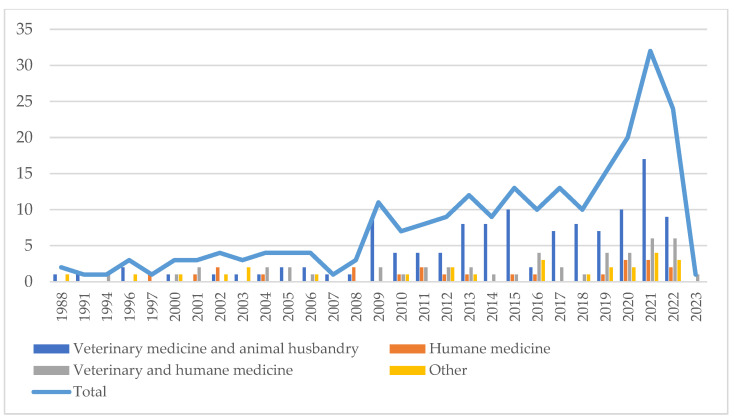
Literature search and general characteristics of the included studies.

**Figure 3 pharmaceutics-15-02326-f003:**
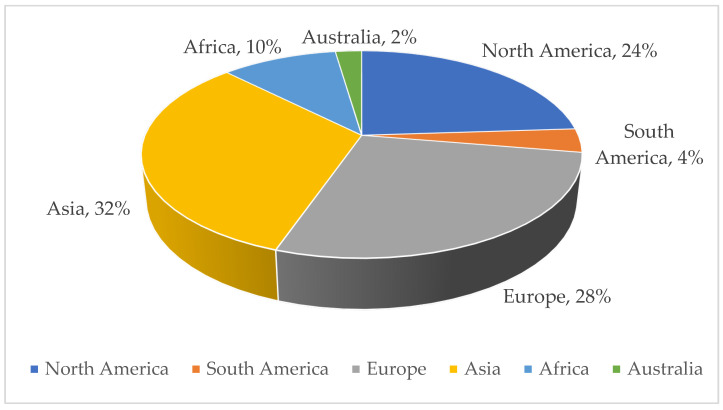
Geographical origin of the sample.

**Figure 4 pharmaceutics-15-02326-f004:**
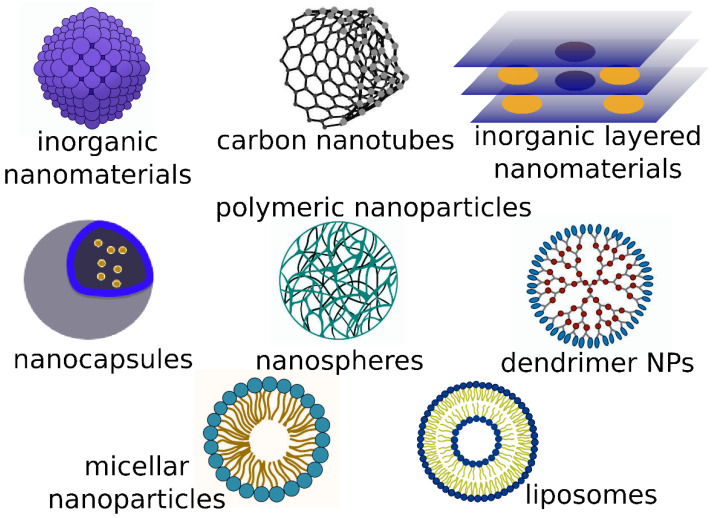
Types of nanoparticles used in veterinary medicine and livestock husbandry.

**Figure 5 pharmaceutics-15-02326-f005:**
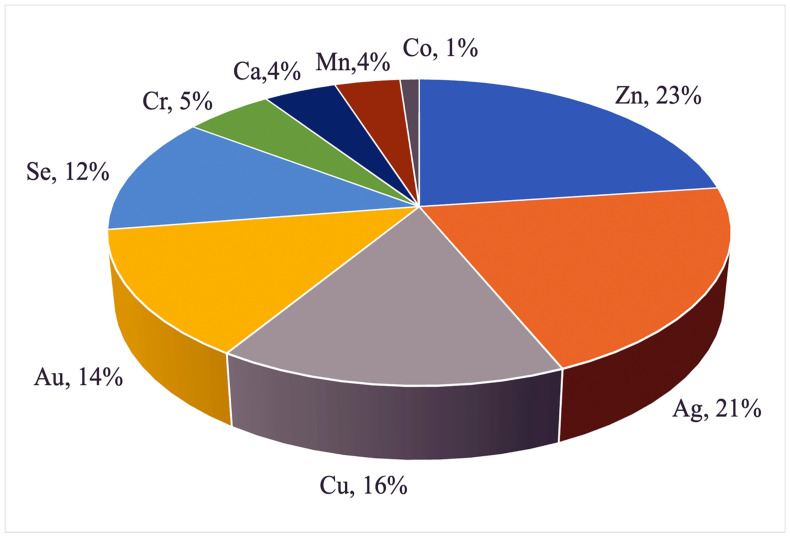
Percentage of the use of metal nanoparticles in animal husbandry.

**Figure 6 pharmaceutics-15-02326-f006:**
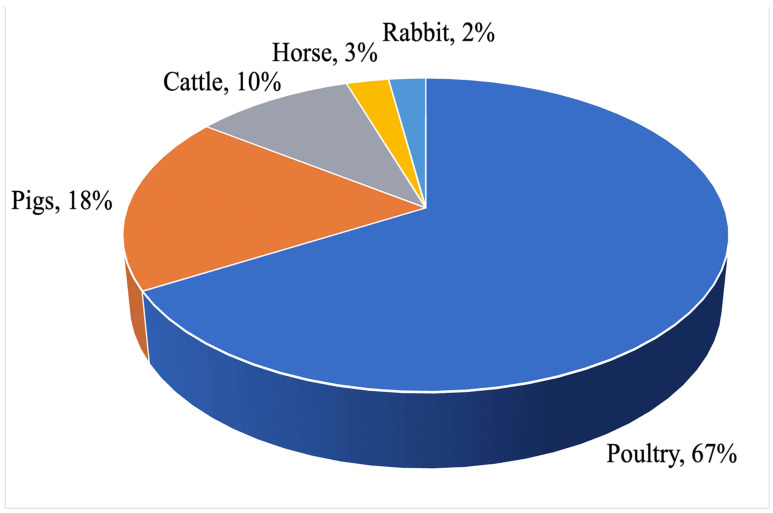
Application of metal nanoparticles in animal species (%).

**Table 2 pharmaceutics-15-02326-t002:** Nanoparticles used for respiratory delivery of veterinary vaccines and their effectiveness.

NP Type	Composition	Antigen/Species/Delivery Route	Efficacy/Ref.
Polymeric	PLG—PVAl microparticle, and 60% NP mix	*Toxoplasma gondii*; Tachyzoite protein extract: SAG1; Cholera toxin (CT)/Ovine (sheep)/NAS	Systemic and local immune response. Consistent and higher IgA in nasal secretions and serum than soluble antigen./[171].
	PLGA	Bovine parainfluenza 3 virus (BPI3V) proteins/Dairy calves (bovine)/NAS	Enhanced and sustained mucosal IgA response compared to i.n. modified live virus commercial vaccine./[172]
	Chitosan	Inactivated NDV/Broiler chicken, layer hens/NAS	Increased IgA humoral response in layers, not broilers./[173]
	CS spray-dried microparticle	BLSOmp31/Ovine (sheep)/NAS	Induced local and systemic immune response in sheep, biphasic release of antigen from microsphere./[174]
	Fungal chitosan	Foot and mouth disease whole virus/Guinea pig/NAS	Higher IgG production in comparison to vaccination with virus alone./[175]
Liposome	PC (zwittterionic); PS (−ve) or Stearylamine (SA) (+ve)	Formalin-inactivated NDV/SPF Leghorn chicken/NAS	PC induced the highest secretory IgA and systemic humoral responses. LPS co-administration increased vaccine efficacy./[176]
	Hydrogenated soybean phospholipids	Inactivated APEC strain KAI-2, O-78/SPF chicken/Coarse spray OU	Reduction in the number of challenged bacteria and clinical signs was observed in chickens after a challenge with APEC./[177]
Liposome–mucoadhesive polymer	PC (zwitterionic) and tremella or xanthan gum	Inactivated influenza H5N3/SPF Leghorn chicken/NAS	Mucoadhesive liposome vesicles induced higher immune response than the virus alone and liposome without the polymer./[178]
Montanide™ IMS adjuvant NP	Not disclosed	Live IBV/SPF chicken (also commercially used in all farm animals)/i.n.	Better than non-adjuvanted vaccine and montanide oil-in-water emulsion; i.n. administration is better than coarse spray./[179]
Adenovirus	BAdV-3	BHV-1 glycoprotein gD, BRSV IL-6/Bovine (cattle)/i.n.	Induces antigen-specific immune responses./[180]
ISCOMs	Quil A saponin	BHV-1 viral membrane proteins/Calves/i.m.	Better protection than commercial attenuated vaccine and higher antibody response produced./[181]

## Data Availability

Not applicable.

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
