# Peer review of "Meeting Contemporary Challenges: Development of Nanomaterials for Veterinary Medicine"

_pharmaceutics, 2023, doi:10.3390/pharmaceutics15092326_

Round 1

Reviewer 1 Report

Minor comments:Line 51- The lower range of the size of the cells doesn't seem to be correct. It should be closer to 1um. Line 183-186 The sentence is ambiguous and unclear. Line 186- The term "advanced research" is not meaningful. Some sentences seem to be incomplete or missing key information, making them difficult to understand. For example, in Line 195, "The effectiveness of the use of antibacterial drugs enclosed in liposomes in the treatment of infectious diseases, in particular, bovine mastitis [56,57], against gram-positive and negative bacteria [44,58], against multiresistant bacterial infections [59], against Mycobacterium avium [60]."

Some information has been repeated. For instance, the effectiveness of silver nanoparticles (AgNPs) against antibiotic-resistant bacteria is mentioned multiple times.    The sentence in Line 212 that starts with “Silver nanoparticles (AgNPs)…” can be removed.    AgNPs has been spelled out multiple times on Line 216 and Line 212.    Line 252: The sentence that begins with "stem cell-based cell therapies..." should start with a capital "S" in "Stem," as it's a new sentence.    Line 255: There appears to be a typographical error with "regenareation." It should be "regeneration."    Line 293- "Spontaneous tumours in cats and dogs have been proposed as the best animal models for human cancer, and thus have been used in preclinical studies for the development of new drugs and imaging probes [95]." There seems to be a missing "for" in "...studies for the development...    Line 303: "Humane oncology" - this might be a mistake or confusion in the terminology. "Human oncology" seems to be the more appropriate term here. The same applies to Line 351.    Line 349- The section "Summarising the above, nanoformulations …” could use a bit of clarification. Do you mean it is difficult to differentiate between the advancements and techniques used in human and veterinary nanotechnological approaches to cancer treatment?    The legend of Figure 3 is not accurate and does not represent the correlation.    Figure 4. The data visualization is not appropriate for the data. A bar graph with the number of publications as the y-axis to convey the information would be more suitable.     Major comments:    Technical Challenges: The complexity of gene therapy— including the delivery of genes into cells, control of gene expression, and the potential for off-target effects— poses significant technical challenges. These hurdles need to be effectively addressed for the successful application of gene therapy in both veterinary and human medicine.    

Grammar needs major improvement.

Author Response

Thank you for reviewing the manuscript.

Please, find below the responses (in italics) to your comments (normal text).  

Minor comments:  

Line 51- The lower range of the size of the cells doesn't seem to be correct. It should be closer to 1um.

Thank you. It was a typo, corrected.  

Line 183-186 The sentence is ambiguous and unclear.

The growing pace of development of antibiotic resistance and the duration of the creation process, as well as the introduction of new generations of antimicrobial drugs, the search for new ways to increase the effectiveness and reduce the toxicity of existing dosage forms is promising.  

The whole paragraph was revised.  

Line 186- The term "advanced research" is not meaningful.  

The whole paragraph was revised, as mentioned above.  

Some sentences seem to be incomplete or missing key information, making them difficult to understand. For example, in Line 195, "The effectiveness of the use of antibacterial drugs enclosed in liposomes in the treatment of infectious diseases, in particular, bovine mastitis [56, 57], against gram-positive and negative bacteria [44, 58], against multiresistant bacterial infections [59], against Mycobacterium avium [60]."  

The sentence was revised.  

Some information has been repeated. For instance, the effectiveness of silver nanoparticles (AgNPs) against antibiotic-resistant bacteria is mentioned multiple times. The sentence in Line 212 that starts with “Silver nanoparticles (AgNPs)…” can be removed.

AgNPs has been spelled out multiple times on Line 216 and Line 212. The whole paragraph was revised. The properties of AgNPs are described in different contexts, there is no redundancy.  

Line 252: The sentence that begins with "stem cell-based cell therapies..." should start with a capital "S" in "Stem," as it's a new sentence.

Done  

Line 255: There appears to be a typographical error with "regenareation." It should be "regeneration."

Corrected  

Line 293- "Spontaneous tumours in cats and dogs have been proposed as the best animal models for human cancer, and thus have been used in preclinical studies for the development of new drugs and imaging probes [95]." There seems to be a missing "for" in "...studies for the development...

Done  

Line 303: "Humane oncology" - this might be a mistake or confusion in the terminology. "Human oncology" seems to be the more appropriate term here. The same applies to Line 351.

Done  

Line 349- The section "Summarising the above, nanoformulations …” could use a bit of clarification. Do you mean it is difficult to differentiate between the advancements and techniques used in human and veterinary nanotechnological approaches to cancer treatment?

Exactly, we followed the Reviewer’s suggestion  

The legend of Figure 3 is not accurate and does not represent the correlation.

The caption of the Figure 3 was corrected.  

Figure 4. The data visualization is not appropriate for the data. A bar graph with the number of publications as the y-axis to convey the information would be more suitable.

The Figure 4 shows percentage of publications related to species. Therefore, a pie chart is adequate for presentation of such data.  

Major comments:   Technical Challenges: The complexity of gene therapy— including the delivery of genes into cells, control of gene expression, and the potential for off-target effects— poses significant technical challenges. These hurdles need to be effectively addressed for the successful application of gene therapy in both veterinary and human medicine.

The corresponding paragraph was added to the end of the 2.2.5 section of the manuscript.  

Grammar needs improvement.

Done  

Reviewer 2 Report

This review topic is of interest although this topic was published recently in 2019 https://www.ncbi.nlm.nih.gov/pmc/articles/PMC6968591/. The Pharmaceutics Journal readers are well versed in much of the nanoparticle dogma reviewed in the manuscript. Much of the examples are for human medicine development and not veterinary. For the examples in veterinary health it would be good to know whether this work is development or has made it to market. It would be good to have more comment on the differences and challenges of developing Nano formulations for veterinary application versus human. For example, veterinary medicines have to be low cost, stable for on farm application (poor cold chain), regulations on environmental impact, the differences in clinical development and registration. Veterinary products also have different profiles between application in companion animal and food -producing animals that should be reviewed. Section 2.2.3 Tissue scaffolds appears no relationship to veterinary, when would this be used? Is there a real market? Figure 2 is poor quality images.

This review topic is of interest although this topic was published recently in 2019 https://www.ncbi.nlm.nih.gov/pmc/articles/PMC6968591/. The Pharmaceutics Journal readers are well versed in much of the nanoparticle dogma reviewed in the manuscript. Much of the examples are for human medicine development and not veterinary. For the examples in veterinary health it would be good to know whether this work is development or has made it to market. It would be good to have more comment on the differences and challenges of developing Nano formulations for veterinary application versus human. For example, veterinary medicines have to be low cost, stable for on farm application (poor cold chain), regulations on environmental impact, the differences in clinical development and registration. Veterinary products also have different profiles between application in companion animal and food -producing animals that should be reviewed. Section 2.2.3 Tissue scaffolds appears no relationship to veterinary, when would this be used? Is there a real market? Figure 2 is poor quality images.

Author Response

Thank you for reviewing the manuscript.

Please, find below the responses (in italics) to your comments (normal text).

This review topic is of interest although this topic was published recently in 2019 https://www.ncbi.nlm.nih.gov/pmc/articles/PMC6968591/. The Pharmaceutics Journal readers are well versed in much of the nanoparticle dogma reviewed in the manuscript. Much of the examples are for human medicine development and not veterinary. For the examples in veterinary health, it would be good to know whether this work is development or has made it to market. It would be good to have more comment on the differences and challenges of developing Nano formulations for veterinary application versus human. For example, veterinary medicines have to be low cost, stable for on farm application (poor cold chain), regulations on environmental impact, the differences in clinical development and registration. Veterinary products also have different profiles between application in companion animal and food -producing animals that should be reviewed. Section 2.2.3 Tissue scaffolds appears no relationship to veterinary, when would this be used? Is there a real market?

We are aware of the article https://www.ncbi.nlm.nih.gov/pmc/articles/PMC6968591/ (2019). The authors of the article focus on possibilities to use several particular nanoparticles in veterinary medicine.

We did not focus on the nanoparticles themselves, our main focus was on different areas of veterinary medicine and on real examples or possibilities of nanotechnology applications in those areas.

Figure 2 is poor quality images.

The Reviewer, probably judged the figure inserted into the manuscript. The high-quality image is uploaded separately to the Journal.

«The Pharmaceutics Journal readers are well versed in much of the nanoparticle dogma reviewed in the manuscript».

The article is intended not only for chemists or pharmacists, but also for veterinary medicine specialists, most of whom, unfortunately, have only a superficial knowledge of nanotechnology and nanoparticles.

«Much of the examples are for human medicine development and not veterinary.»

Frankly speaking, it is impossible to draw a clear borderline in the application of nanoparticles in veterinary and human medicines. For example, all humane drugs are tested on animals and, upon demonstration of effectiveness, enter the veterinary market.

«It would be good to have more comment on the differences and challenges of developing Nano formulations for veterinary application versus human.»

As mentioned above, there are no differences, all nanomaterials and technologies have been tested on animals. Subsequently, they enter the veterinary market, if this is feasible economically. In other words, the technologies are common, the doses and pharmaceutically active ingredients may be different, while carriers are the same. For example, new generation antibiotics first enter the veterinary market, and only after few years they are released to the human one.

«Section 2.2.3 Tissue scaffolds appears no relationship to veterinary, when would this be used? Is there a real market?»

Yes, it is. This is a fairly popular and promising field of veterinary medicine today, although the main focus may still be on humane medicine. The market is still not wide, since there are many unsolved problems. However, there are great outlooks for using the tissue scaffolds, especially for sport horses and special breed animals.

Reviewer 3 Report

Currently, the effective and successful application of nanomaterials in many fields of veterinary medicine has represented one of the greatest technological developments, where, the findings reported so far have been very promising. In fact, nanoscale materials and forms of drug delivery have been used to now successfully used for the diagnosis, treatment, and prevention of infectious animal diseases and not infectious. The main strength of this manuscript is the presentation of an analysis of the state of the art and the prospects for the application of nanomaterials in veterinary medicine. Similarly, it seems to me that the discussion of a wide range of available nanotechnologies (drugs, diagnostics, feed additives, and vaccines) is of great interest to the reader. Another strength of the manuscript is that it also provides a brief overview of current issues around drug security nanomaterials, the potential risks of their use, and some relevant legal aspects. I thank the authors for such an interesting manuscript, which I have read with great interest and have thoroughly enjoyed reading. However, some points must be addressed to achieve publication quality. I have left some comments hoping that they can help the authors.

General comments

L65: please add a reference.

L73: After the introduction, add a section called methodology. In it, indicate the sites or sources used to search for information, keywords used in the search, inclusion and exclusion criteria taken into account, or even how you classified the articles consulted according to their content. If possible, support your explanation with a figure.

L76: delete the word etc. Please apply this suggestion to your entire manuscript.

L114: I suggest that the authors carry out a deeper discussion about the nanomaterials used in the diagnosis of infectious diseases, placing greater emphasis on the bioanalytical kits used by species. That is, present the findings on this topic by species (cattle, horses, pigs, sheep, goats, dogs, cats, and other species of veterinary interest).

L166: To improve the reader's understanding, it would be very important if examples of drug delivery through systems such as liposomes could be added. I suggest authors add and discuss studies on Bupivacaine micellar, Bupivacaine multilamellar liposomes, polymerized alginate nanoparticles, bupivacaine microcapsules, prilocaine liposomes or cyclodextrin, lidocaine liposomes, lidocaine-prilocaine hybrid nanofilm, tetracaine polymeric nanoparticles, and even in areas such as veterinary ophthalmology with sirolimus liposomes.

L281: the way to refer to the summons is incorrect, please review.

L457 and 497: please remove the phrase that appears between parentheses.

Tables 1 and 2 – Add a table footer that includes an explanation of all abbreviations found in the tables. 

Figures 3 and 4: the information presented is confusing since it does not indicate whether they are proportions or number of publications. In one of them mention is made of a correlation but its expression and explanation in the figure are not clear. I suggest the authors delete these figures since it is information that is already contained in the manuscript. On the contrary, if you want to keep these figures, then they must be modified so that the reader correctly identifies the variables that you want to present.

L545: please add a reference.

L644: Before the conclusions, I suggest the authors add a section called Perspectives and Future directions, where the authors can discuss, for example, the replacement of conventional therapies, the development of more specific therapeutic targets, and the reduction of the incidence of adverse reactions, and even how this technology can contribute in the future to the development of different fields of study in veterinary medicine and other areas of knowledge.

Minor editing of English language required

Author Response

Thank you for reviewing the manuscript.

Please, find below the responses (in italics) to your comments (normal text).

General comments

L65: please add a reference.

Done

L73: After the introduction, add a section called methodology. In it, indicate the sites or sources used to search for information, keywords used in the search, inclusion and exclusion criteria taken into account, or even how you classified the articles consulted according to their content. If possible, support your explanation with a figure.

In our opinion, the methodology section in the review article is redundant, as it does not contain new information, and the studied sources are listed in the bibliography. Additionally, a separate analysis of the processed sources was made (see Figs. 3-4.). Although, if this remark is critical, we could make the corresponding section.

L76: delete the word etc. Please apply this suggestion to your entire manuscript.

Done

L114: I suggest that the authors carry out a deeper discussion about the nanomaterials used in the diagnosis of infectious diseases, placing greater emphasis on the bioanalytical kits used by species. That is, present the findings on this topic by species (cattle, horses, pigs, sheep, goats, dogs, cats, and other species of veterinary interest).

We fully agree with the Reviewer's remark about the importance of using nanomaterials in the diagnosis of infectious diseases of animals. However, ELISA and PCR are currently the main methods of diagnostics have no significant species specificity. In this work, we primarily focused on nanoparticles and nanotechnologies that are directly integrated into veterinary medicine and have contact with a living organism. The description and discussion of nanomaterials used in the diagnosis of infectious diseases deserves a separate article, which we plan to prepare in the nearest future.

L166: To improve the reader's understanding, it would be very important if examples of drug delivery through systems such as liposomes could be added. I suggest authors add and discuss studies on Bupivacaine micellar, Bupivacaine multilamellar liposomes, polymerized alginate nanoparticles, bupivacaine microcapsules, prilocaine liposomes or cyclodextrin, lidocaine liposomes, lidocaine-prilocaine hybrid nanofilm, tetracaine polymeric nanoparticles, and even in areas such as veterinary ophthalmology with sirolimus liposomes.

We agree with the Reviewer on the importance of highlighting examples of liposome drug delivery. However, the addition and detailed discussion of the issues mentioned by the Reviewer will significantly expand the section in question and shift the balance of the description of nanoparticles towards the liposomes. Such an extended discussion could unfairly reduce the importance of other nanoparticles as well as will also require an expansion of their entire manuscript.

Moreover, the application of liposomes in each particular area of veterinary medicine is discussed throughout the sections of the manuscript.

L281: the way to refer to the summons is incorrect, please review.

Checked, everything is correct.

L457 and 497: please remove the phrase that appears between parentheses.

The phrase in parenthesis indicates a desired place for the Tables 1 and 2. This is just an information to the production team for the preparation of the final set-up of the manuscript.

We agree to remove it from the text and to put into a comment instead.

This is a technical note that will not appear in the published material.

Tables 1 and 2 – Add a table footer that includes an explanation of all abbreviations found in the tables. 

All conventional abbreviations are given at the end of the article in the Abbreviations section.

Figures 3 and 4: the information presented is confusing since it does not indicate whether they are proportions or number of publications. In one of them mention is made of a correlation but its expression and explanation in the figure are not clear. I suggest the authors delete these figures since it is information that is already contained in the manuscript. On the contrary, if you want to keep these figures, then they must be modified so that the reader correctly identifies the variables that you want to present.

Yes, as the diagrams are pie charts, they show data in %. The figure captions were corrected.

L545: please add a reference.

 Done

L644: Before the conclusions, I suggest the authors add a section called Perspectives and Future directions, where the authors can discuss, for example, the replacement of conventional therapies, the development of more specific therapeutic targets, and the reduction of the incidence of adverse reactions, and even how this technology can contribute in the future to the development of different fields of study in veterinary medicine and other areas of knowledge.

The corresponding section was added to the manuscript.

Comments on the Quality of English Language

Minor editing of English language required

Done

Reviewer 4 Report

The review article reports the advancement of nanotechnology in the field of veterinary medicine, yet some suggestions should be implemented to improve and update it.

The abstract needs to be re-written and re-phrased to reflect the actual contents of the manuscript

Lines 51-52: What is meant by size compatibility of nanoparticles to cells? Could you elaborate? Also discuss the effects of size variations of nanoparticles in relation to their efficacy. 

Figure 1: it needs to be restructured, for example theranostics mean both diagnosis and treatment, so as a bigger item should be theranostics and from it comes diagnosis and treatment. Also, parasitic infections should be placed under the classification infectious diseases, which should include bacterial, fungal and viral diseases. 

Line 59-60 should be re-phrased

A paragraph about applications of nanoparticles in fish medicine should be added, there are different reviews on this topic and other recent articles on different types of nanoparticles in aquaculture such as: DOI 10.3390/fishes8070333

Another review was published on applications in poultry farms you could take as a reference as well.

Line 182-183: What is meant by this statement? Do you mean Sustainable development goals SDGs?

Line 218: only here it is mentioned one paper about antiviral activity of nanoparticles in veterinary medicine, please check literature and expand the part of antiviral nanoparticles and their mechanisms of action.

The manuscript needs moderate English language editing

Author Response

Thank you for reviewing the manuscript.

Please, find below the responses (in italics) to your comments (normal text).

The review article reports the advancement of nanotechnology in the field of veterinary medicine, yet some suggestions should be implemented to improve and update it.

The abstract needs to be re-written and re-phrased to reflect the actual contents of the manuscript

In our opinion, the abstract fully corresponds to the actual content of the article

Lines 51-52: What is meant by size compatibility of nanoparticles to cells? Could you elaborate? Also discuss the effects of size variations of nanoparticles in relation to their efficacy. 

The sentences mentioned were revised.

Therapeutic efficacy of nanomaterials is a highly complex phenomenon that depends on many different parameters. Size is only one of those. The discussion on “the effects of size variations of nanoparticles in relation to their efficacy” goes far beyond the scope of the present manuscript as well as that of the Special issue.

Figure 1: it needs to be restructured, for example theranostics mean both diagnosis and treatment, so as a bigger item should be theranostics and from it comes diagnosis and treatment. Also, parasitic infections should be placed under the classification infectious diseases, which should include bacterial, fungal and viral diseases. 

We have to disagree with the Reviewer about theranostics, because it is a SIMULTANEOUS diagnosis and treatment. Diagnosis and treatment cannot be derived separately from teranostics. That will not be correct.

We fully agree with the Reviewer that parasitic infections are nosologically accepted to be included in infectious diseases. However, given rather significant differences in the approaches to diagnosis, prevention, and treatment of parasitoses, it is justified and customary to present them as a separate section under the treatment. That is what we did.

Line 59-60 should be re-phrased

Done

A paragraph about applications of nanoparticles in fish medicine should be added, there are different reviews on this topic and other recent articles on different types of nanoparticles in aquaculture such as: DOI 10.3390/fishes8070333

The corresponding discussion was added into the 2.4 section of the manuscript.

Another review was published on applications in poultry farms you could take as a reference as well.

Thank you for the comment. The manuscript describes comprehensively the state-of-the-art for different areas of veterinary medicine.

Line 182-183: What is meant by this statement? Do you mean Sustainable development goals SDGs?

The corresponding paragraph has been revised.

Line 218: only here it is mentioned one paper about antiviral activity of nanoparticles in veterinary medicine, please check literature and expand the part of antiviral nanoparticles and their mechanisms of action.

The paragraph was revised.

Comments on the Quality of English Language

The manuscript needs moderate English language editing

Done

Round 2

Reviewer 3 Report

I thank the authors for considering my comments in the first revision of their manuscript. It seems to me that the article has improved substantially, so I am convinced that if this review material is published, readers will have an important contribution to favor the development of veterinary medicine, especially in the field of nanotechnology.

However, in the new manuscript, I have found some aspects that require the authors' attention before publishing the article.

First, although this article is not a systematic review, it should be considered that when conducting scientific research (in this case under a review article modality), a search and information analysis methodology must be followed in accordance with the proposed title and aim. Therefore, in general, it is feasible that this manuscript can be added to a section called methodology in which the sites or sources used to search for information, keywords used in the search, inclusion and exclusion criteria taken into account, or even how you classified the articles consulted according to their content.

In this journal, you will find examples of articles published under the review article format that consider a section of these characteristics.

In this reviewer's opinion, the use of nanomaterials in the diagnosis of infectious diseases in animals is a topic that could interest readers and what I consider should be discussed. Even though the authors have clarified that they have considered this topic for a future publication, I think it is convenient to add some paragraphs that allow emphasizing bioanalysis kits used in animals. Therefore, my suggestion to the authors is to present the most outstanding findings on this topic by species (cattle, horses, pigs, sheep, goats, dogs, cats, and other species of veterinary interest).

In Figures 3 and 4, the percentages of each item presented must be added.

Finally, section 5 (L654-683) has no references. Please add those that you used for your writing.

Author Response

We would like to thank the Reviewer for her/his time spent to revise our manuscript. Please, find below the answers to the Reviewer's comments.

First, although this article is not a systematic review, it should be considered that when conducting scientific research (in this case under a review article modality), a search and information analysis methodology must be followed in accordance with the proposed title and aim. Therefore, in general, it is feasible that this manuscript can be added to a section called methodology in which the sites or sources used to search for information, keywords used in the search, inclusion and exclusion criteria taken into account, or even how you classified the articles consulted according to their content.

In this journal, you will find examples of articles published under the review article format that consider a section of these characteristics.

Response

We have taken into account the reviewer's comments and added the Methodology section (section 2)

In this reviewer's opinion, the use of nanomaterials in the diagnosis of infectious diseases in animals is a topic that could interest readers and what I consider should be discussed. Even though the authors have clarified that they have considered this topic for a future publication, I think it is convenient to add some paragraphs that allow emphasizing bioanalysis kits used in animals. Therefore, my suggestion to the authors is to present the most outstanding findings on this topic by species (cattle, horses, pigs, sheep, goats, dogs, cats, and other species of veterinary interest).

Response

We cannot agree with the Reviewer’s opinion since Diagnostics is not within the scope of the present Special Issue dedicated to Biodegradable Nanomaterials for Targeted Drug Delivery. We inserted that section ONLY because there is a huge number of papers devoted to the topic, moreover a part of them are of theranostic interest. But still, the application of nanomaterials for diagnostics is beyond the scope of both the present review and the Special Issue. We are going to prepare a separate review on the topic.

In Figures 3 and 4, the percentages of each item presented must be added.

Response

Percentages were added. After insertion of the Methodology section, the numbers of the figures in question became 5 and 6.

Finally, section 5 (L654-683) has no references. Please add those that you used for your writing.

Response

The Reviewer's recommendation was taken into account. The references were added.

Reviewer 4 Report

The previous comments were not sufficiently addressed.

The manuscript still needs English editing

Author Response

We would like to thank the Reviewer for her/his time spent to revise our manuscript. Please, find below the answers to the Reviewer's comments.

The previous comments were not sufficiently addressed.

Response

We revised the Reviewer’s comments once again and added some paragraphs related to the application of nanomaterials in fish treatment and aquaculture as well as their effects on the environment and humans.

The publication suggested by the Reviewer was added together with 11 other publications on the topic.

Other comments received in the previous round were considered in full.

The manuscript still needs English editing.

Response

Done